# Adjuvanted-SARS-CoV-2 Spike Protein-Based Microparticulate Vaccine Delivered by Dissolving Microneedles Induces Humoral, Mucosal, and Cellular Immune Responses in Mice

**DOI:** 10.3390/ph16081131

**Published:** 2023-08-10

**Authors:** Smital Patil, Sharon Vijayanand, Ipshita Menon, Keegan Braz Gomes, Akanksha Kale, Priyal Bagwe, Shadi Yacoub, Mohammad N. Uddin, Martin J. D’Souza

**Affiliations:** Center for Drug Delivery and Research, Vaccine Nanotechnology Laboratory, College of Pharmacy, Mercer University, Atlanta, GA 30341, USA; smitalrajan.patil@live.mercer.edu (S.P.); sharon.c.p.vijayanand@live.mercer.edu (S.V.); ipshita.jayaprakash.menon@live.mercer.edu (I.M.); keegan1717@gmail.com (K.B.G.); akanksha.madhav.kale@live.mercer.edu (A.K.); priyal.bagwe@live.mercer.edu (P.B.); shadi.yacoub@live.mercer.edu (S.Y.)

**Keywords:** adjuvanted vaccine, microparticles, microneedles, spike protein, COVID-19, subunit vaccine, immune response, humoral response, cellular responses

## Abstract

COVID-19 continues to cause an increase in the number of cases and deaths worldwide. Due to the ever-mutating nature of the virus, frequent vaccination against COVID-19 is anticipated. Most of the approved SARS-CoV-2 vaccines are administered using the conventional intramuscular route, causing vaccine hesitancy. Thus, there is a need for an effective, non-invasive vaccination strategy against COVID-19. This study evaluated the synergistic effects of a subunit microparticulate vaccine delivered using microneedles. The microparticles encapsulated a highly immunogenic subunit protein of the SARS-CoV-2 virus, such as the spike protein’s receptor binding domain (RBD). Adjuvants were also incorporated to enhance the spike RBD-specific immune response. Our vaccination study reveals that a microneedle-based vaccine delivering these microparticles induced spike RBD-specific IgM, IgG, IgG1, IgG2a, and IgA antibodies. The vaccine also generated high levels of CD4+ and CD8a+ molecules in the secondary lymphoid organs. Overall, dissolving microneedles delivery spike RBD antigen in microparticulate form induced a robust immune response, paving the way for an alternative self-administrable, non-invasive vaccination strategy against COVID-19.

## 1. Introduction

Coronavirus disease (COVID-19), caused by the severe acute respiratory syndrome coronavirus 2 (SARS-CoV-2) [1], continues to drive a global rise in cases, with more than 687 million cases and more than 6.8 million deaths globally as of May 2023 [2]. It has been reported that 70.3% of the world population received at least one dose of a COVID-19 vaccine [3]. In contrast, only 32.2% of the population in low-income countries has received one dose of a COVID-19 vaccine [3]. Many factors, including inefficient vaccine rollout, poor infrastructure, need for cold chain storage, vaccine hesitancy, etc., contribute to the lower vaccination rates [4,5,6]. Additionally, needle fear, complacency, and lack of confidence in vaccines are some of the reasons that contribute to vaccine hesitancy, one of the top ten global health threats in 2019, according to the World Health Organization (WHO) [5,7]. Therefore, there is a need for a COVID-19 vaccine that overcomes these challenges, including avoiding cold-chain storage and having a more compliant, needle-free vaccination strategy.

The causal agent of COVID-19, SARS-CoV-2, belongs to the *Coronaviridae* family. It consists of four structural proteins: spike protein (S), nucleoprotein (N), envelope protein (E), and membrane protein (M). The S protein comprises two functional subunits, S1 and S2 [8], and is the most exposed protein to the SARS-CoV-2 virus [9]. The receptor binding domain (RBD) protein, part of the S1 subunit, is critical for binding to the angiotensin-converting enzyme-2 (ACE-2) receptors on the host cell. In humans, this receptor is abundant throughout the body, in the epithelial layer of the lungs, kidneys, brain, and blood vessels [10,11]. Therefore, the RBD of the spike protein (Spike RBD) is a promising antigen for vaccine development as the spike RBD is critical in mediating viral entry in the host [8,12]. According to previous studies, the RBD from the S1 subunit protein is highly immunogenic and has been proven to elicit highly neutralizing antibodies and robust humoral immune responses in the convalescent serum of individuals infected by SARS-CoV-2 [13]. Thus, we utilized the spike RBD of the SARS-CoV-2 virus as an antigen candidate for testing a subunit microparticulate vaccine.

Microparticles (MPs) formulated using biodegradable polymers such as poly(lactic-co-glycolic acid) (PLGA) have been studied as systems for the delivery of vaccine antigens [14,15,16,17], for soluble antigens like spike RBD, factors such as short half-life, low molecular size, and poor cross-presentation by activation of mainly the MHC-II pathway contribute towards the low immunogenic potential of the soluble antigen [16,17,18,19,20,21]. In contrast, encapsulating the antigen helps protect the antigen and increases cellular uptake by antigen-presenting cells (APCs) [22,23,24]. This leads to effective cross-presentation of the antigen by the MHC-I pathway, which helps induce helper and cytotoxic T-cell responses [19]. Activation of the MHC-I pathway is essential for presenting the antigen to CD8+ cytotoxic T cells and clearance of viral infection [25]. Additionally, particulate-based vaccines have other immunological advantages, such as effective circulation of the antigen to draining lymph nodes, activation of germinal centers, and efficient antigen presentation by follicular dendritic and helper T cells [17,18,26,27]. In this study, the spike RBD vaccine antigen was encapsulated in a PLGA polymer matrix to form a microparticulate vaccine using the double emulsion method. Formulation of MPs with PLGA polymer offers excellent biodegradability and biocompatibility properties, thereby reducing cytotoxicity [20,28]. Moreover, the double emulsion formulation method has the advantage of preserving the biological activity of the protein, as in the single emulsification technique, the hydrophilic agents such as proteins or peptides can easily diffuse in the aqueous phase [20,29].

Adjuvants have been used along with vaccine antigens to enhance vaccine-induced immunity by activating APCs [30]. Alhydrogel^®^ and AddaVax™ have been approved for use in marketed vaccines such as HPV (human papillomavirus), Gardasil 9, and Fluad, respectively [31]. Additionally, TLR agonists such as CpG-based adjuvants have recently been approved for use in marketed vaccines such as Heplisav-B [31]. Each adjuvant has a distinct mechanism for enhancing vaccine-induced immunity. Alhydrogel^®^ induces a Th2-biased immune response by enhancing uptake by APCs to activate helper T cells [32]. AddaVax™ activates Th1 along with Th2 immune responses by activating APCs to stimulate CD4+ and CD8+ T cells [33]. CpG 7909 is a synthetic oligonucleotide that induces a Th1-biased immune response by activating CD4+ T cells to produce a robust antibody and mucosal response [34,35]. In previous studies, we have tested the effectiveness of adjuvants in the microparticulate form [14,36]. We observed that the microparticulate form of Alhydrogel^®^ and AddaVax™ were able to enhance antigen-specific humoral and cellular immune responses [14]. In this study, we tested the effect of CpG 7909 along with Alhydrogel^®^ and AddaVax™ to enhance immune responses against the spike RBD in particulate form.

Current vaccines against COVID-19 are administered using the painful intramuscular route of administration, which necessitates the exploration of non-invasive alternative immunization routes such as microneedle-based delivery of vaccines intradermally [37]. Skin harbors abundant specialized dendritic cells such as Langerhans cells (LCs) and dermal dendritic cells. These immune cells can be targeted using microneedles (MN) vaccine delivery techniques to induce a robust immune response [38,39]. Additionally, MN is minimally invasive as they do not affect the nerve endings, leading to a painless delivery method [40,41]. This vaccine approach also allows the potential for self-administration, easing the burden on healthcare professionals. Self-administration can accelerate mass immunization, especially during a global pandemic [40,42].

Our previous “proof-of-concept” study demonstrated that a subunit microparticulate vaccine with the SARS-CoV spike glycoprotein (GP) as a model antigen was administered using MN-induced spike GP-specific serum antibody and cellular responses in spleens of immunized mice [14]. Accordingly, in this follow-up study, we next tested the efficacy of an MN-based vaccine with spike RBD MPs with and without adjuvants. Our vaccine-delivering spike RBD MPs using MN with and without adjuvant MPs elicited humoral and cellular immune responses in vaccinated mice.

## 2. Results

### 2.1. Formulation and Characterization of Spike RBD, Adjuvant Microparticles and Dissolvable Microneedles

The MPs were prepared using the double emulsion technique and then lyophilized, resulting in a percentage recovery yield ranging from 88% to 91%. The particle size average varied, with the Spike RBD MPs, Alhydrogel^®^ MPs, AddaVax™ MPs, and CpG 7909 having sizes of 0.7 µm, 1.19 µm, 1.05 µm, and 0.9 µm, respectively. The PDI of the MPs fell into the range of 0.55–0.67, while the zeta potential measurements ranged between −20 mV and −25 mV. The total antigen content in the Spike RBD MPs was evaluated to be 89.94 ± 5.23%. Furthermore, SEM images demonstrated that the Spike RBD MPs were spherical. The MN was successfully formulated using a spin-casting method for the spike RBD suspension, spike RBD MPs with and without adjuvants. SEM analysis confirmed the consistent dimensions and shape of the formed MN. The MNs were able to penetrate the paraffin^®^ skin model up to 4 layers without deformation or breaking, indicating that the MNs were mechanically strong and can penetrate up to 300–400 µm in depth. The MNs were also able to penetrate the excised mouse skin, forming micropores that were seen clearly with the help of methylene blue staining solution (Figure 1).

### 2.2. Spike RBD Specific Antibodies Detected in Sera of Vaccinated Mice

Two weeks after each dose and during sacrifice, we obtained mice serum samples and tested for spike RBD-specific IgM, IgG, IgA, IgG1, and IgG2a antibodies. After the prime dose, the IgM levels were significantly higher in mice vaccinated with MN-based vaccine than in the naïve group. The IgM levels decreased significantly in vaccinated mice after week 2 and remained insignificant until week 10 (Figure 2).

The total IgG antibodies were significantly higher in mice immunized with MN delivering spike RBD in suspension and microparticulate form in comparison to naïve after the first dose and throughout the end of the study. Similarly, the total IgG responses were elevated in mice immunized with the spike RBD MPs and adjuvant MPs in comparison to naïve. Total IgG responses when compared with the (spike RBD suspension) MN group were significantly high in the following groups: (spike RBD MPs + Alhydrogel^®^ MPs), (spike RBD MPs + Alhydrogel^®^ MPs + AddaVax™ MPs) MN, and (spike RBD MPs + Alhydrogel^®^ MPs + AddaVax™ MPs + CpG 7909 MPs) MN, after the second booster dose and continued to be high until the terminal week (Figure 3).

Serum IgA levels were significantly high for all groups after the prime dose and elevated in the following weeks after booster doses when compared to naïve. The serum IgA levels were significantly high in mice immunized with (spike RBD MPs + Alhydrogel^®^ MPs + AddaVax™ MPs + CpG 7909 MPs) MN group as compared to the group that received spike RBD suspension in MN (Figure 4).

We also tested for subtypes of IgG: IgG1 and IgG2a. Our results revealed that the IgG1 levels were significantly higher in (spike RBD MPs + Alhydrogel^®^ MPs + AddaVax™ MPs) MN and (spike RBD MPs + Alhydrogel^®^ MPs + AddaVax™ MPs + CpG 7909 MPs) MN in comparison with naïve following the prime dose and booster doses and remained significant until terminal week. There was a significant difference in the IgG1 levels in mice immunized with (spike RBD MPs + Alhydrogel^®^ MPs + AddaVax™ MPs + CpG 7909 MPs) MN as compared to (spike RBD suspension) MN group after the first booster dose and remained significant until terminal week (Figure 5).

Our results indicated that serum IgG2a levels were significantly high in the following groups after each dose: (spike RBD suspension)MN, (spike RBD MPs + Alhydrogel^®^ MPs) MN, (spike RBD MPs + Alhydrogel^®^ MPs + AddaVax™ MPs) MN as compared to naïve. Serum IgG2a was significantly higher in the terminal week for the (spike RBD MPs + Alhydrogel^®^ MPs + AddaVax™ MPs + CpG 7909 MPs) MN compared to naïve. The IgG2a response was significantly high as compared to (spike RBD suspension)MN during the terminal week for the following groups: (spike RBD MPs + Alhydrogel^®^ MPs + AddaVax™ MPs) MN and (spike RBD MPs + Alhydrogel^®^ MPs + AddaVax™ MPs + CpG 7909 MPs) MN (Figure 6).

### 2.3. Spike RBD Specific IgA Antibodies Detected in Lung Homogenates of Vaccinated Mice

The IgA levels in lung homogenate samples were determined using ELISA. The IgA levels were significantly high in lung homogenates of mice vaccinated with the following groups: (spike RBD suspension) MN, (spike RBD MPs + Alhydrogel^®^ MPs) MN, (spike RBD MPs + Alhydrogel^®^ MPs + AddaVax™ MPs) MN, (spike RBD MPs + Alhydrogel^®^ MPs + AddaVax™ MPs+ CpG 7909 MPs) MN in comparison with naïve. IgA antibodies were significantly high in (spike RBD MPs + Alhydrogel^®^ MPs + AddaVax™ MPs+ CpG 7909 MPs) MN vaccine group in comparison with (spike RBD suspension) MN group (Figure 7).

### 2.4. Elevated Percentage of Cells Expressing CD4 and CD8a Molecules in Spleen of Vaccinated Mice

In spleenocytes, the percentage of cells expressing CD4 and CD8a molecules was significantly high in the following groups: Spike RBD suspension MN, Spike RBD MPs MN, (Spike RBD MPs + Alhydrogel^®^ MPs) MN, (spike RBD MPs + Alhydrogel^®^ MPs + AddaVax™ MPs) MN, (spike RBD MPs + Alhydrogel^®^ MPs + AddaVax™ MPs+ CpG 7909 MPs) MN in comparison with naïve. The responses were equivalent when spike RBD suspension groups were compared with spike RBD MPs with and without adjuvants (Figure 8).

### 2.5. Elevated Percentage of Cells Expressing CD4 and CD8a Molecules in Lymph Nodes of Vaccinated Mice

In lymphocytes, the percentage of cells expressing CD4 and CD8a molecules was significantly high in the following groups: Spike RBD suspension MN, Spike RBD MPs MN, (Spike RBD MPs + Alhydrogel^®^ MPs) MN, (Spike RBD MPs + Alhydrogel^®^ MPs + AddaVax™ MPs) MN, (Spike RBD MPs + Alhydrogel^®^ MPs + AddaVax™ MPs+ CpG 7909 MPs) MN in comparison with naïve. In groups (Spike RBD MPs + Alhydrogel^®^ MPs + AddaVax™ MPs) MN, (Spike RBD MPs + Alhydrogel^®^ MPs + AddaVax™ MPs+ CpG 7909 MPs) MN, the percentage of CD4 molecules was significantly high as compared to Spike RBD Suspension MN group. The percentage of cells expressing CD8a molecules was significantly high in (Spike RBD MPs + Alhydrogel^®^ MPs + AddaVax™ MPs+ CpG 7909 MPs) MN groups in comparison to (spike RBD suspension) MN and naïve group (Figure 9).

## 3. Discussion

This study demonstrated immune responses induced by MN-based delivery of spike RBD MPs with and without adjuvants. The MN-based delivery system of subunit microparticulate vaccine with and without adjuvants induced a strong spike in RBD-specific antibodies and cellular immune responses in immunized mice.

PLGA MPs protect the antigen from enzymatic degradation and other extracellular agents [43]. As soluble antigens are relatively less immunogenic due to poor cross-presentation than particulate vaccines [18,20], we formulated spike RBD MPs using PLGA. To enhance vaccine-induced immunity of spike RBD MPs, we prepared adjuvant MPs similarly. MPs in the 0.1–3 µm range have previously been shown to enhance antigen uptake [44,45]. The size of the MPs in this study ranged from 0.7 to 1.2 µm, which likely contributed to enhanced uptake by APCs, resulting in enhanced immunogenicity. The surface charge of the MPs was tested and ranged from −20 to −25 mV. The change in the zeta potential could be attributed to the different antigens or adjuvants encapsulated in the MPs. Higher negative or positive zeta potential is also an indication of less agglomeration [46] of MPs. The antigen content of spike RBD in the MPs was more than 89%. Formulating MPs using a double emulsion solvent evaporation method followed by lyophilization did not affect the antigen content of spike RBD in MPs to a great extent. Thus, this formulation method effectively loads soluble protein antigens to formulate effective microparticulate carrier systems. These results were consistent with our previously published studies [14].

The administration of the MPs-loaded MN vaccine to mice allowed delivery of the antigen and adjuvant MPs into the dermal layers of the skin to be easily taken up by APCs, producing a strong immune response. In contrast, administration of a spike RBD suspension was less immunogenic than microparticulate vaccine, as indicated by the antibody and cellular responses. Thus, administration of the vaccine antigen in microparticulate form instead of suspension form using microneedles is advantageous from an immunological perspective, as the vaccine MPs can be better taken up by antigen-presenting cells. The study of MN delivery of vaccine antigen against SARS-CoV-2 has been in preclinical testing. However, this study reports the effects of the integration of microparticulate vaccine delivered via MN [14,40,47]. A phase 1 clinical trial was conducted for the “PittCoVacc” vaccine, which administered the spike protein suspension in an MN-based vaccine [48,49].

The presence of antigen-specific binding antibodies is indicative of humoral response, essential for the elimination of the pathogen via increased phagocytosis by complement-mediated lysis or opsonization [50]. Therefore, after administration of MN with spike RBD suspension, spike RBD MPs, and spike RBD MPs with adjuvant MPs, we evaluated the presence of binding antibodies. We found that IgM levels peaked after the prime dose for mice vaccinated with MN, delivering spike RBD MPs with and without adjuvants. In the following weeks, the IgM levels significantly decreased. When tested for IgG and IgA levels, the antibodies were significantly high after each dose but increased after booster doses and continued to remain high until the terminal week in mice immunized with spike RBD MPs with and without adjuvants. Studies have reported that upon vaccination, IgM antibodies are produced in the initial week, followed by isotype class switching to IgG and IgA [51]. Our results indicated similar isotype class switching of spike RBD-specific IgM to spike RBD-specific IgG and IgA antibodies in immunized mice as compared to non-immunized mice. Studies have shown that serum IgG and IgA antibodies are vital in binding and neutralizing the SARS-CoV-2 virus [52]. Specifically, IgA is more potent than IgG in neutralizing [53]. Thus, we also tested for IgA antibodies, specifically in the lungs. Our data demonstrated that mice vaccinated with spike RBD MPs with and without adjuvants using the MN route produced high levels of IgA antibodies. The presence of significantly high levels of IgG and IgA antibodies indicates our vaccine delivery system produces a humoral immune response. Although binding antibodies are crucial, the presence of neutralizing antibodies can validate vaccine efficacy. Our follow-up studies aim to address the neutralizing capacity of these antibodies in the vaccinated mice to validate the vaccine’s effectiveness.

Further, subtyping of IgG antibodies for IgG1 and IgG2a responses revealed the presence of adjuvant MPs in polarizing the antibody responses. IgG1 antibodies are indicative of Th2-biased immune response, and IgG2a antibodies are indicative of Th1-biased immune response [54,55]. An increase in both IgG1 and IgG2a antibody levels was observed in mice vaccinated with spike RBD MPs with and without adjuvants. A Th1-biased immune response was observed in mice immunized with MN delivering spike RBD MPs + Alhydrogel^®^ MP including and not including AddaVax™ MPs and CpG 7909 MPs. A Th2-biased immune response was demonstrated by mice vaccinated with spike RBD MPs + Alhydrogel^®^ MPs + AddaVax™ MPs with and without CpG 7909 MPs. The antibody responses produced by spike RBD suspension in MN were only Th1-biased. Therefore, utilizing the microparticulate form of the antigen and adjuvants in combination elicited elevated levels of IgG2a and IgG1, indicating Th1 and Th2 immune responses in vaccinated mice.

Antigen presentation via the MHC molecules on the APCs determines which T cells are activated [25]. Helper T cells (CD4+ T cells) are activated when the antigen is presented on MHC II molecules. These helper T cells stimulate B cells to produce antibodies. On the other hand, cytotoxic T cells (CD8+ T cells) are activated when the antigen is presented on MHC I molecules [56]. Helper T cells and cytotoxic T cells indicate Th2- and Th1-biased immune responses, respectively. The presence of helper T cells is crucial for producing antibody responses to neutralize the virus before cell entry and cytotoxic T cells, which are vital to target and eliminate virus-infected cells. We found that the percentage of cells expressing CD4 and CD8a molecules were significantly higher in mice immunized with spike RBD MPs including and not including adjuvants than in unvaccinated mice. We tested these in both spleen and lymph nodes specific for spike RBD. The mice vaccinated with spike RBD suspension demonstrated helper T cells only in the lymph nodes but no cytotoxic T cell response in either spleens or lymph nodes. The data suggested that both Th1 and Th2 immune responses were seen in vaccinated mice with MPs compared to unvaccinated mice. Both these immune responses are essential for the effectiveness of the vaccine. 

The rapid development in the vaccines against COVID-19 led to the approval of over two dozen vaccine candidates globally against COVID-19 [57]. These vaccines have certainly demonstrated effectiveness to either prevent infection or prevent the progression of severe diseases, reducing hospitalizations and mortality significantly [58]. According to the Institute of Health metrics and evaluation, the effectiveness of the current SARS-CoV-2 vaccines ranges from 66%–94% [58]. One of the limitations of this study is the evaluation of the existing vaccine to compare the effectiveness of the vaccine candidates with the existing vaccines. Affordability of COVID-19 vaccines in low- and middle-income countries is challenging due to the limited accessibility of the vaccines, vaccine awareness, and the limited resources to manufacture vaccines in their countries [59]. Moreover, while negotiating with vaccine manufacturers, their purchasing power is lower as compared to high-income countries [60]. These are some of the key factors contributing to the access and distribution of vaccines in low- and middle-income countries [59,60]. Efforts to enable equitable access to vaccines regardless of the economic status of the countries are crucial to increasing vaccination rates in low- and middle-income countries [59,61].

Overall, this study demonstrated the immunogenic potential using the induction of spike RBD-specific antibodies and cellular responses produced by our MN-based subunit microparticulate vaccine. Our follow-up studies will include testing for the neutralizing capacity of the antibodies using a pseudovirus neutralization assay. The presence of memory markers is crucial for understanding the durability of the vaccine formulation. Thus, follow-up studies will include memory responses observed in the B and T cells of the spleen and lymph nodes. Additionally, follow-up studies will include stability studies for the vaccine candidates.

## 4. Materials and Methods

### 4.1. Materials

Vaccine antigen, the spike receptor binding domain, was procured from BEI Resources, NIAID (National Institute of Allergy and Infectious Diseases), NIH (National Institutes of Health): Spike Glycoprotein Receptor Binding Domain (RBD) from SARS-Related Coronavirus 2, Wuhan-Hu-1 with C-Terminal Histidine Tag, Recombinant from HEK293T Cells (NR-52946). Sodium hyaluronate (MW 100 kDa) was obtained from Lifecore Biomedical (Chaska, MN, USA). Poly(lactic-co-glycolic) acid (PLGA) 75:25 (Resomer^®^ RG 752H) was obtained from Evonik Industries (Birmingham, AL, USA). Dichloromethane (DCM) was acquired from Fischer Scientific (Hampton, NH, USA). Span^®^ 80 and trehalose dihydrate were purchased from Millipore Sigma (Burlington, MA, USA). 3-(4,5-Dimethylthiazol-2-yl)-2,5-diphenyltetrazolium bromide (MTT) and Pierce Micro BCA Assay Kit was obtained from Thermo Fischer (Waltham, MA, USA). The 8 × 8 array polydimethylsiloxane (PDMS) MN molds were purchased from Micro Point Technologies (Singapore). Murine dendritic cells (DC 2.4) were offered as a kind gift by Dr. Kenneth L. Rock (Dana-Farber Cancer Institute Inc., Boston, MA, USA). Adjuvants such as Alhydrogel^®^ (alum) and AddaVax™ (MF59-like adjuvant) were purchased from InvivoGen (San Diego, CA, USA). Horseradish peroxidase (HRP)-conjugated goat anti-mouse secondary IgM, IgG, IgA, IgG1, and IgG2a antibodies were acquired from Invitrogen (Rockford, IL, USA). 3,3′,5,5′-tetramethylbenzidine (TMB) was purchased from Becton, Dickinson & Co. (Franklin Laks, NJ, USA). Six-to-eight-week-old mice (Swiss-webster strain) were obtained from Charles River Laboratories (Wilmington, MA, USA). Cell culture supplies such as Dulbecco’s Modified Eagle’s Medium (DMEM), trypsin EDTA solution, fetal bovine serum (FBS), and penicillin/streptomycin were acquired from American Type Culture Collection (Manassas, VA, USA).

### 4.2. Methods

#### 4.2.1. Preparation and Characterization of Spike RBD Microparticles along with Adjuvant Microparticles

The subunit vaccine MPs and adjuvant MPs were formulated with the spike receptor binding domain (RBD) as the vaccine antigen and Alhydrogel^®^, AddaVax™, and CpG 7909 as vaccine adjuvants. The spike RBD antigen and adjuvant MPs were prepared using a double emulsion technique with PLGA as the polymer, as previously described [14,15,62,63]. In brief, the MPs were prepared by formulating a double emulsion (w/o/w) using antigen or adjuvant (aqueous phase) and PLGA in DCM (organic phase) and span 80 as a primary emulsifier formulating the primary emulsion (w/o). Further, this primary emulsion was incorporated into the PVA solution (aqueous phase) to form a double emulsion (w/o/w). The emulsions were developed by using a probe homogenizer at 17,000 rpm. The formulated double emulsion was placed on a magnetic stirrer for 5 h to allow for the evaporation of DCM solvent. Next, the emulsion was centrifuged for 15 min at 15,000 rpm at 4 °C to obtain a pellet of MPs. This pellet was resuspended in 2% *w*/*v* trehalose and placed in the Labconco™ benchtop freeze dryer to acquire a lyophilized powder. The RBD vaccine and adjuvant MPs were further characterized for various parameters. The size of MPs, zeta potential, and polydispersity index were evaluated using a Malvern Zeta Sizer. The MPs were viewed using a scanning electron microscope (SEM) using the method described previously [14]. The percentage of antigen content in spike RBD MPs was measured using the Micro Bicinchoninic acid (BCA) assay according to the instructions from the manufacturer. The percentage of antigen content was measured as encapsulation efficiency using the following formula:$$\mathrm{Encapsulation}\;\mathrm{Efficiency}=\frac{Experimental\;protein\;content\;in\;MPs}{Theoretical\;protein\;content\;in\;MPs}\times 100$$

The recovery yield in percent of lyophilized MPs was calculated with the following formula:$$\mathrm{Recovery}\;\mathrm{yield}\;(\%)=\frac{Weight\;of\;lyophilized\;microparticles}{Weight\;of\;solid\;ingredients\;in\;the\;formulation\;pre-lyophilization}\times 100$$

#### 4.2.2. Formulation of Dissolvable Microneedle-Loaded with Vaccine and Adjuvant Microparticles

Microneedles were formulated using the spin-casting method previously established in our laboratory [14,15,47,62,64]. First, depending on the group, the corresponding RBD suspension, RBD MPs, and adjuvant MPs were added to the trehalose solution. The RBD suspension in MN groups was formulated by adding the appropriate dose in the trehalose solution and was further formed in a hydrogel polymer by adding sodium hyaluronate. This hydrogel was weighed carefully to be 25 mg each in 8 × 8 array PDMS molds. The PDMS mold was centrifuged at 270 g at 15 °C for 15 min and then dried in a vacuum desiccator. Then, a backing layer of concentrated sodium hyaluronate was added to the molds and allowed to dry overnight. The final formulation was taken out from the PDMS molds by applying 3M tape and placed on a supportive base to form a formulation similar to a band-aid for application. Similarly, formulations for other treatment groups were formulated by adding the RBD MPs and adjuvant MPs (Alhydrogel^®^ MPs, AddaVax™ MPs, and CpG 7909 MPs) corresponding to the treatment group mentioned in Table 1. The morphology of the MN before and after application to mouse skin was studied under the SEM. The MNs were placed on a stub covered with double-sided adhesive 12 mm carbon tabs and visualized using Phenom™ Benchtop SEM, Nanoscience instruments, AZ. The mechanical strength of the MNs was assessed by testing the ability of MNs to penetrate a validated skin model formed of Parafilm M^®^ and excised mouse skin. The formulated MN patch was applied to the excised mouse skin, followed by staining with 1% methylene blue solution.

In this table, Column 1 represents the groups in which the mice were categorized to be vaccinated, Column 2 represents the treatment the mice received in the vaccine and column 3 represents the route of vaccination was through intradermal route.

#### 4.2.3. In Vivo Immunization with Dissolvable Microneedles Loaded with Vaccine and Adjuvant Microparticles

The efficacy of the MN-based vaccine was tested on Six-to-eight-week-old (Swiss Webster strain) mice procured from Charles River Laboratories, Wilmington, MA, USA. The vaccination study was performed in compliance with the approved Mercer University IACUC protocol #A2004006. The groups were vaccinated according to the corresponding treatment group, as mentioned in Table 1. The vaccination regiment included one prime and two booster doses at weeks 0, 3, and 6, respectively. The mice were bled by the tail-snip method bi-weekly after each dose to obtain mice sera, which was tested for antibodies. At week 10, the mice were sacrificed, and their lungs, spleen, and lymph nodes were harvested for further analysis. Prior to every dose, around 2 sq.cm of mouse fur on the dorsal region was cleared out with depilatory cream to ensure easy administration of the MN.

#### 4.2.4. Quantification of Spike RBD Specific Antibody Responses in Serum

After each dose, serum samples were collected biweekly at weeks 2, 5, and 8 and during sacrifice at week 10 (terminal week). The blood samples were centrifuged at the following parameters: 1700 g, 10 min, 4 °C to acquire serum and preserved at −80 °C for further analysis. Serum samples were tested for the following antibodies using enzyme-linked immunosorbent assay (ELISA): IgM, IgG, IgG1, IgG2a, and IgA. First, spike RBD protein (500 ng/well) was coated on high-binding 96-well Microlon^®^ plates and left at 4 °C overnight. For 3 h at 37 °C, the wells were blocked with a blocking buffer 3% bovine serum albumin (BSA). The wells received serum samples diluted to 1:50 with 1% *w*/*v* BSA before incubating at 4 °C overnight. The next day, secondary goat anti-mouse IgM, IgG, IgG1, IgG2a, and IgA antibodies that had been conjugated with horseradish peroxidase were added to the wells in the range of 1:2000–1:5000, depending on the antibodies tested and placed to incubate at 37 °C for 1.5 h. TMB substrate was added to the wells first, and then color-stopping solution of 0.1 M H_2_SO_4_ was added to the well in 1:1 ratio. The plate was then read at 450 nm using a plate reader. The plate was washed with 0.05% *w*/*v* PBS solution thrice before every step. 

#### 4.2.5. Measurement of Spike RBD Specific IgA Responses in Lung Homogenates

The lung samples were collected post-sacrifice and processed into lung homogenates. To measure Spike-RBD-protein-specific IgA titers, ELISAs were conducted on the lung homogenate samples. Initially, 96-well plates of a high-binding nature were covered with Spike RBD protein (500 ng/well) and placed overnight at 4 °C. 3% BSA was added as a blocking agent and incubated for 3 h at 37 °C. Diluted lung homogenate samples with 1% *w*/*v* BSA to 1:100 were added to the wells and allowed to incubate overnight at 4 °C. Next day, horseradish-peroxidase-conjugated secondary antibodies-goat anti-mouse IgA was added to the wells at 1:3000 dilution and allowed to incubate (1.5 h at 37 °C). Subsequently, 50 µL of TMB substrate was added to each well and quenched by adding 50 µL 0.1 M H_2_SO_4_. The plate was then read at 450 nm. Prior to every step, 0.05% *w*/*v* PBS-Tween solution was used to wash the wells thrice.

#### 4.2.6. Evaluation of Spike RBD Specific Helper and Cytotoxic T Cell Responses in Spleen and Lymph Nodes

The mice were sacrificed during week 10, and spleen samples were isolated. Single-cell suspension was obtained using a 40 µm cell strainer. Using ammonium chloride potassium (ACK) lysis buffer, the samples were treated to lyse red blood cells (RBCs). Similarly, lymph nodes (inguinal and brachial) were also processed into single-cell suspensions except for the RBC lysis step. The splenocytes and lymphocytes were resuspended in media (70% FBS, 25% DMEM, and 5% DMSO), followed by storing at −80 °C for further analysis of T cell expression. CD4+ and CD8+ T cells present in spleens and lymph nodes were evaluated using flow cytometry. The cells were first thawed on ice, centrifuged at the specified parameters below, and resuspended in 1 mL complete DMEM media to eliminate DMSO from the media used for preserving the cells at −80 °C. Followed by plating the cells (24-well plate) overnight and stimulated with 2.5 ng/well Interlukein-2 (IL-2) overnight. The IL-2 was removed post-centrifugation, and 5 μg of spike RBD protein was added for 24 h. After that, they were washed and stained with APC-labeled, FITC-labeled CD4 and CD8 anti-mouse antibodies, respectively. Following incubation, the samples were washed three times using PBS and evaluated with flow cytometry to measure the percentage of T cells expressing CD4+ and CD8+. Hence, to test the specificity of the CD4+ and CD8+ T cells toward SARS-CoV-2, the splenocytes and lymphocytes were stimulated in vitro with the spike RBD antigen. Before all these steps, the cells were centrifuged at the following parameters: 150 g, 8 min, 4 °C to remove media, and cells were resuspended using fresh DMEM.

### 4.3. Statistical Analysis

GraphPad Prism version 9.2.0 for Windows (GraphPad Software, San Diego, CA, USA) was utilized to perform all statistical analyses. One-way Analysis of Variance (ANOVA) and Two-way ANOVA were used to examine comparisons between multiple groups, and post hoc analysis was conducted using Tukey’s test. The data obtained are presented as mean ± SEM. Statistical significance was denoted by *p*-values lower than 0.05. Significance was determined using a scale where ns denoted non-significant, (*) represented statistical significance at *p* < 0.05, (**) *p* < 0.01, (***), *p* < 0.001, and (****) *p* < 0.0001.

## 5. Conclusions

In this study, spike RBD microparticles and adjuvant microparticles were formulated and loaded in dissolvable microneedles to form a microneedle-based microparticulate vaccine. In vivo testing revealed that the microneedles delivering spike RBD microparticles enhanced antibody responses compared to spike RBD suspension in microneedles. This microneedle-based microparticulate vaccine with and without adjuvants induced Th1 and Th2 mediated immune responses indicated by the presence of IgG1, IgG2a antibodies, and helper, cytotoxic T cells. Therefore, this study establishes the synergistic immunogenic effect of a microparticulate vaccine delivered via microneedles. This study lays the foundation for a prospective intradermal vaccination strategy using a subunit microparticulate vaccine against COVID-19.

## Figures and Tables

**Figure 1 pharmaceuticals-16-01131-f001:**
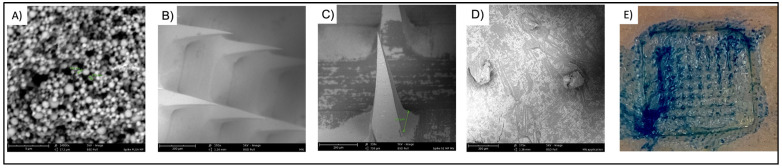
Scanning electron microscopic images of (**A**) spike RBD microparticles (14,000×) showing microparticle size of around 600 nm (**B**) microneedle array from a latitudinal view (**C**) single microneedle demonstrating base length of 112 mm (**D**) dissolved microneedles post-vaccination (175×) (**E**) Digital microscopic image of methylene blue stained micropores created by microneedles in excised mouse skin.

**Figure 2 pharmaceuticals-16-01131-f002:**
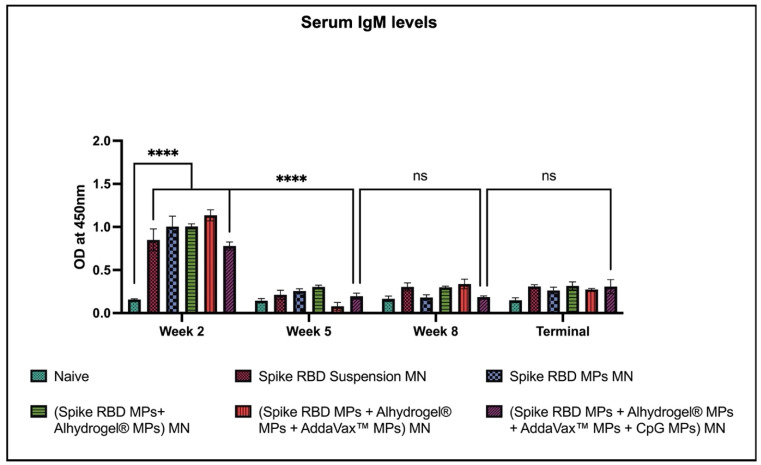
Spike RBD protein-specific-antibody levels in serum for IgM were measured in immunized and non-immunized mice. ELISA was used to analyze serum samples at dilution 1:50. Data are expressed as mean ± SEM. Two-way ANOVA followed by post hoc Tukey’s multiple comparison test was used to analyze data: **** *p* < 0.0001.

**Figure 3 pharmaceuticals-16-01131-f003:**
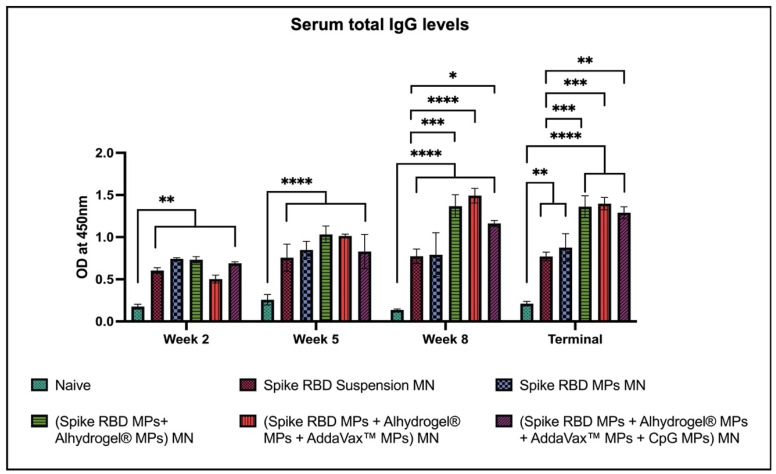
Spike RBD protein-specific-antibody levels in serum for IgG were measured in immunized and non-immunized mice. ELISA was used to analyze serum samples at dilution 1:50. Data are expressed as mean ± SEM. Two-way ANOVA followed by post hoc Tukey’s multiple comparison test was used to analyze data: * *p* < 0.05, ** *p* < 0.01, *** *p* < 0.001, **** *p* < 0.0001.

**Figure 4 pharmaceuticals-16-01131-f004:**
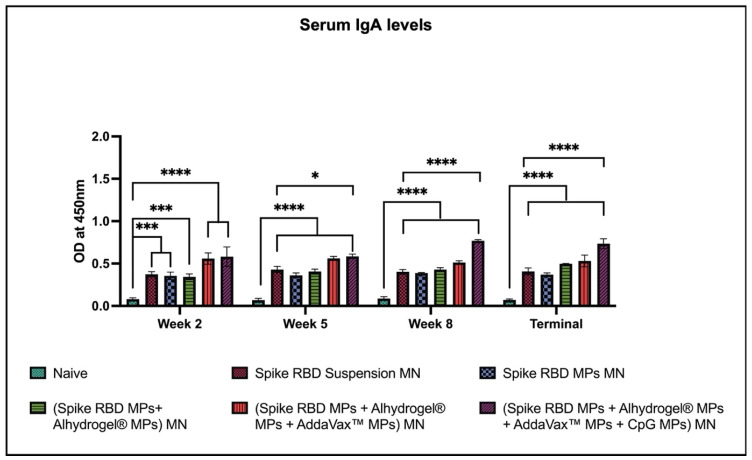
Spike RBD protein-specific-antibody levels in serum for IgA were measured in immunized and non-immunized mice. ELISA was used to analyze serum samples at dilution 1:50. Data are expressed as mean ± SEM. Two-way ANOVA followed by post hoc Tukey’s multiple comparison test was used to analyze data: * *p* < 0.05, *** *p* < 0.001, **** *p* < 0.0001.

**Figure 5 pharmaceuticals-16-01131-f005:**
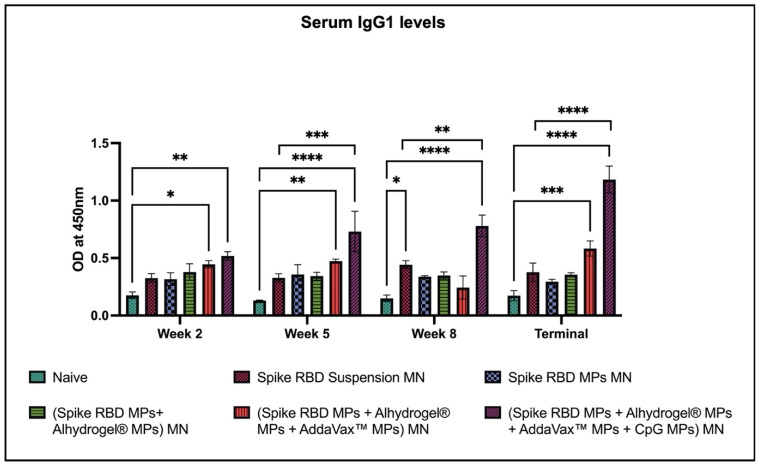
Spike RBD protein-specific-antibody levels in serum for IgG1 were measured in immunized and non-immunized mice. ELISA was used to analyze serum samples at dilution 1:50. Data are expressed as mean ± SEM. Two-way ANOVA followed by post hoc Tukey’s multiple comparison test was used to analyze data: ** p* < 0.05, ** *p* < 0.01, *** *p* < 0.001, **** *p* < 0.0001.

**Figure 6 pharmaceuticals-16-01131-f006:**
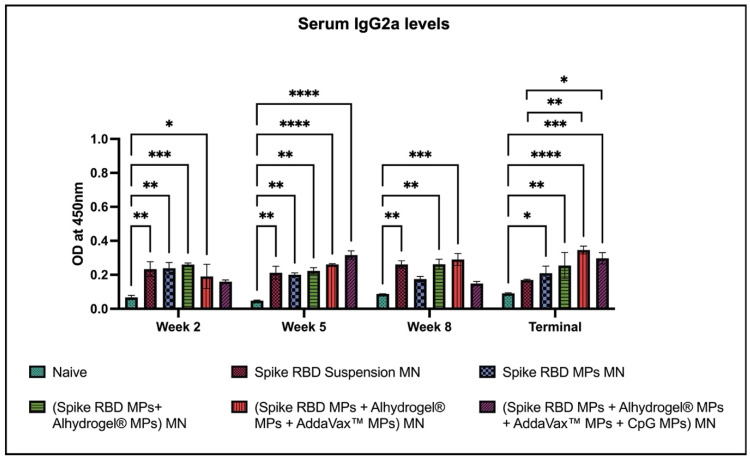
Spike RBD protein-specific-antibody levels in serum for IgG2a were measured in immunized and non-immunized mice. ELISA was used to analyze serum samples at dilution 1:50. Data are expressed as mean ± SEM. Two-way ANOVA followed by post hoc Tukey’s multiple comparison test was used to analyze data: * *p* < 0.05, ** *p* < 0.01, *** *p* < 0.001, **** *p* < 0.0001.

**Figure 7 pharmaceuticals-16-01131-f007:**
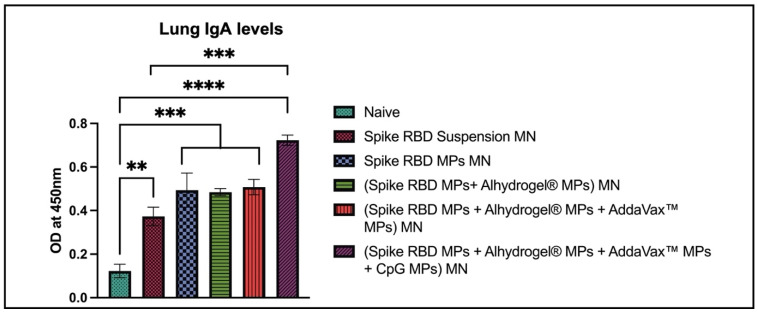
Spike RBD protein-specific-antibody levels in lung homogenates for IgA were measured in immunized and non-immunized mice. ELISA was used to analyze samples at dilution 1:100. Data are expressed as mean ± SEM. Two-way ANOVA followed by post hoc Tukey’s multiple comparison test was used to analyze data: ** *p* < 0.01, *** *p* < 0.001, **** *p* < 0.0001.

**Figure 8 pharmaceuticals-16-01131-f008:**
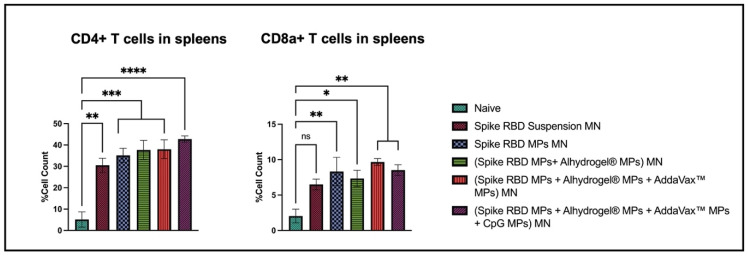
Percent of CD4+ and CD8+ cells in spleens of immunized and non-immunized mice analyzed using flow cytometry. Data are expressed as mean ± SEM. One-way ANOVA followed by post hoc Tukey’s multiple comparison test was used to analyze data: * *p* < 0.05, ** *p* < 0.01, *** *p* < 0.001, and **** *p* < 0.0001.

**Figure 9 pharmaceuticals-16-01131-f009:**
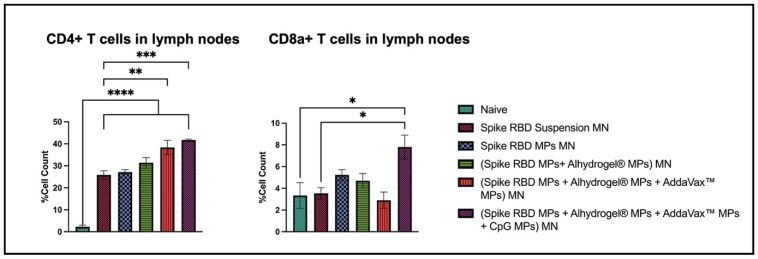
Percent of CD4+ and CD8+ cells in lymph nodes of immunized and non-immunized mice analyzed using flow cytometry. Data are expressed as mean ± SEM. One-way ANOVA followed by post hoc Tukey’s multiple comparison test was used to analyze data: * *p* < 0.05, ** *p* < 0.01, *** *p* < 0.001, and **** *p* < 0.0001.

**Table 1 pharmaceuticals-16-01131-t001:** Description of vaccination study groups.

Vaccine Group	Description of Vaccine Received	Route of Vaccination
Naïve	No treatment	-
entry 2	data	data
Spike RBD Suspension MN	Spike RBD suspension only in MNs	Intradermal
Spike RBD MPs MN	Spike RBD PLGA MPs in MNs	Intradermal
(Spike RBD MPs + Alhyrogel^®^ MPs + AddaVax™) MN	Spike RBD PLGA MPs, Alhyrogel^®^ PLGA MPs and Addavax™ PLGA MPs in MN	Intradermal

## Data Availability

The data presented in this study is available upon request from the corresponding author.

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
