# Peer review of "Adjuvanted-SARS-CoV-2 Spike Protein-Based Microparticulate Vaccine Delivered by Dissolving Microneedles Induces Humoral, Mucosal, and Cellular Immune Responses in Mice"

_pharmaceuticals, 2023, doi:10.3390/ph16081131_

Round 1
Reviewer 1 Report
- Mechanical characterisation is missing. Please, add mechanical strength test, insertion test, as well as microscopy analysis.
- What is the difference between this work and https://doi.org/10.1016/j.ijpharm.2023.122583?
Reviewer 2 Report
Dear authors,
This is a very valuable study, providing very interesting results, expectably with application in the near future. It was grateful to read this manuscript.
The manuscript is generally clear and well-written.
Just few comments concerning aspects that can still be improved:
Line 41: Coronaviridae family
Lines 267-269: the final part of this sentence should be rephrased.
Lines 390, 416, 460: please indicate centrifugal force (g), instead of unclear rpm values.
Line 397: “under the SEM”. It was expected additional information in a M&M section.
Lines 450-451: “The cells were first thawed on ice and suspended in DMEM to eliminate DMSO”. Unclear procedure, please reformulate: DMEM volume? “…to eliminate DMSO?” Was DMSO eliminated?
The manuscript is generally clear and well-written.
Reviewer 3 Report
Interesting article on a microneedle adjuvant system using SARS-Cov-2 RBD as an antigen model.
PLGA microparticles are indeed a very promising system for antigen delivery in vaccine formulations. However, the stability of the antigen in this type of system is a matter of great concern and needs to be carefully evaluated. If the authors have already carried out this type of study, it is advisable that they put at least a few sentences about it.
Encapsulation was evaluated by total protein dosage, but encapsulation itself is already somewhat aggressive for proteins. I would like to know if the process influenced the immunogenicity of the protein. It did not eliminate it, given the good results presented in the in vivo tests, but to what extent was there a loss?
Reviewer 4 Report
The authors have done remarkable work by studying the synergistic effects of a subunit microparticulate vaccine delivered using microneedles.
However, some issues have been highlighted in the PDF file, and comments have been made using sticky notes.
The efficacy and adverse effects of the vaccine candidates can't be ignored.
Discussion about the efficacy of the existing vaccines and affordability by low and middle-income countries should be incorporated.

Round 2
Reviewer 1 Report
The manuscript entitled: "Immune responses induced by Adjuvanted-Spike Protein-based Microparticulate Vaccine delivered by Dissolving Microneedles against SARS-CoV-2" is similar to previous work from the group with the same authors and a similar title in two other different journals.
https://www.ncbi.nlm.nih.gov/pmc/articles/PMC9503342/
https://www.sciencedirect.com/science/article/pii/S0378517323000030?via%3Dihub
The authors did not include any mechanical characterisation for microneedles, even after required. This item is an essential criterion for product development to guarantee microneedles penetration in the skin.
